# Cholesterol in Class C GPCRs: Role, Relevance, and Localization

**DOI:** 10.3390/membranes13030301

**Published:** 2023-03-03

**Authors:** Ugochi H. Isu, Shadi A Badiee, Ehsaneh Khodadadi, Mahmoud Moradi

**Affiliations:** Department of Chemistry and Biochemistry, University of Arkansas, Fayetteville, AR 72701, USA

**Keywords:** G-protein-coupled-receptors, GPCRs, membrane protein, protein–lipid interactions, sterols, cholesterol, class C GPCRs

## Abstract

G-protein coupled receptors (GPCRs), one of the largest superfamilies of cell-surface receptors, are heptahelical integral membrane proteins that play critical roles in virtually every organ system. G-protein-coupled receptors operate in membranes rich in cholesterol, with an imbalance in cholesterol level within the vicinity of GPCR transmembrane domains affecting the structure and/or function of many GPCRs, a phenomenon that has been linked to several diseases. These effects of cholesterol could result in indirect changes by altering the mechanical properties of the lipid environment or direct changes by binding to specific sites on the protein. There are a number of studies and reviews on how cholesterol modulates class A GPCRs; however, this area of study is yet to be explored for class C GPCRs, which are characterized by a large extracellular region and often form constitutive dimers. This review highlights specific sites of interaction, functions, and structural dynamics involved in the cholesterol recognition of the class C GPCRs. We summarize recent data from some typical family members to explain the effects of membrane cholesterol on the structural features and functions of class C GPCRs and speculate on their corresponding therapeutic potential.

## 1. Introduction

Many recent studies are geared towards deciphering the structures of G-protein coupled receptors (GPCRs) through several methods, most commonly crystallography and cryogenic electron microscopy (Cryo-EM). This is simply because many medications are designed to target GPCRs due to their central role in many biological functions. All GPCRs in a lipid bilayer are not stand-alone molecules, rather they interact with other components of the bilayer e.g., lipids and sterols, most notably cholesterols [1]. Some members of GPCRs exist and function as monomers, especially within the class A GPCRs, while other GPCRs, mostly the class C GPCRs, form dimers [2,3], and oligomers with themselves or other GPCRs [4]. Most recently, several determined GPCR structures often existing as dimers, appear to indicate the presence of cholesterol. A notable example is the 2-adrenergic receptor (2AR), a class A GPCR crystallized with cholesterol molecules and a component of the dimer interface consisting of the addition of post-translational palmitate groups from each protomer [1]. GPCRs function within cholesterol-rich membranes and an insufficient or excessive amount of cholesterol within the membrane could induce conformational changes in many GPCRs which would result in various diseases [5,6]. These effects of cholesterol could result in indirect changes by altering the mechanical properties of the lipid environment or direct changes by binding to specific sites on the protein [7,8,9]. There are a number of studies and reviews on how cholesterol modulates class A GPCRs, but this area of study is yet to be fully explored for class C GPCRs. Recent advances in experimental and computational power have enabled researchers to investigate the role of lipids in various membranes and solvable proteins, at the atomic level using molecular dynamics simulation [10,11,12,13,14,15,16,17,18].

Class C GPCRs consist of about 23 receptors with a unique characterization compared to other GPCR classes and exist as obligate homo- (e.g., CaS) [19,20,21,22] or hetero- (e.g., GABA${}_{B}$) dimers [23,24,25,26]. They are made up of three unique structural elements: a seven-transmembrane domain which is responsible for allosteric ligand recognition but is uniquely dimeric in the case of class C GPCRS [27]; an unusually large extracellular venus flytrap domain (VFT) which has a double-lobed structure with a crevice between them to serve as the orthosteric binding site; and a cysteine-rich domain (CRD) that links the VFT region to the 7TM region (Figure 1) [3]. However, some class C GPCRs, like GABA${}_{B}$ receptors, lack the cysteine-rich domain [24,25,28,29,30]. Due to these distinct structural features and mandatory dimerization, the class C GPCRs have been the most complex of the GPCRs in terms of understanding their activation mechanism [31,32,33,34,35]. Using several methods such as crystallization [30], lipid cubic phase [36], and most commonly single particle Cryo-EM, structures of over 20 human class C GPCRs have been solved to date [37], comprising metabotropic glutamate receptors (mGluR1–5,mGluR7) [36,38,39,40,41,42,43,44,45,46], gamma-aminobutyric acid receptors (GABA${}_{B}$1 and GABA${}_{B}$2) [23,24,47], calcium-sensing receptors (CaS) [48,49,50], the extra-cellular domain of taste receptors (TAS1R1–TAS1R3) [51,52,53,54,55], and orphan receptors (GPR158, GPR179, GPR156) [56,57,58,59,60]. Similarly to other GPCR structures, class C GPCR structures are solved with inclusion of cholesterol or cholesteryl hemisuccinate (CHS) to the detergent mix during crystallization and recently, Cryo-EM (Table 1). However, some of the structures also have bound cholesterol or cholesteryl hemisuccinate acting as ligands to the already determined structures (Table 1). The argument for cholesterol addition varies from stabilizing the protein to aiding dimerization. Experimental analysis and, most recently, molecular dynamics simulations [16,17,61,62,63,64] have been used to decipher the possible role of cholesterol in these protein structures. In this review, we will discuss the relevance and position of cholesterol molecules in class C GPCR structures and functions.

### 1.1. Cholesterol–Membrane Interactions

The plasma membrane of eukaryotic cells consists of various lipids displaying high biochemical variability in both their apolar moiety and their polar head [67,68]. Sterols are a class of lipids that are a key component of the plasma membrane and are characterized by their steroid hydrocarbon ring structure. One specific sterol, cholesterol, makes up a vital part of the plasma membrane of eukaryotic cells. Cholesterol is crucial for membrane dynamics and organization [69,70,71] and it is also necessary for viability and cell proliferation [72]. The structural features of cholesterol qualify it to interact with proteins and other membrane lipids in several different ways through a variety of different interaction domains [67]. Cholesterol consists of a tetracyclic fused ring skeleton with a single hydroxyl group, a double bond, and a flexible iso-octyl hydrocarbon sidechain [73] which allows it to take on a wide array of conformations [67]. The hydroxyl group is said to contribute significantly to the amphiphilic behavior of cholesterol, causing it to orient in membranes [74]. It is also essential in the hydrogen bond formation between cholesterol and water [75], as well as other lipid membranes in the cell [76]. The hydroxyl group can form two distinct types of hydrogen bonds (acceptor and donor) with a polar group belonging to either a membrane lipid or a protein. Cholesterol is able to affect the physical behavior and dynamics of the cell membrane by interacting with membranes rich in sphingolipids such as lipid “rafts” [77,78,79], or by being present in the liquid disordered (Ld) phase of membranes which contain a large number of glycerophospholipids such as phosphatidylcholine [80]. As a result, cholesterol can alter the properties and dynamics of proteins in the membrane [73,81,82,83]. When bound to cholesterol, some integral membrane proteins could become activated or inactivated [84,85,86]. In recent times, there has been considerable interest in cholesterol interaction sites in membrane proteins. Certain proteins function in cholesterol-rich domains, while others have direct interactions with cholesterol through their transmembrane domains, and sometimes act as ligands [11,61]. The more common cholesterol binding sites in membrane proteins include the cholesterol recognition/interaction amino acid consensus (CRAC)/CARC domain [87], the cholesterol consensus motif (CCM) [88,89] and the sterol-sensing domain (SSD) [90,91]. All of these listed are structural features in proteins that could result in preferential involvement with cholesterol. Several studies have shown that protein–cholesterol interactions are more common in proteins with sequences comprising of the CRAC motif [92], a short peptide segment at the tail of a transmembrane helix comprising of 5–13 amino acid residues. The CRAC motif consists of a well defined linear sequence of amino acids [67,92,93,94,95,96,97] identified via the following pattern: a leucine or valine residue, 1–5 non-specific amino acid residues, tyrosine, another 1–5 residues of any amino acid, and finally a lysine or arginine residue [-L/V-(X)1–5-Y-(X)1–5-R/K-, with (X)1–5 representing between one and five residues of any amino acid] [92,93,94,95]. Rhodopsin, the β(2)-adrenergic receptor, and the serotonin(1A) receptor are examples of GPCRs that have been identified with the CRAC motif recognition site [92]. The major difference between the CARC and CRAC motif is that one exhibits a preference for the outer membrane leaflet (CARC), while the mirror sequence (CRAC) is located in the inner membrane leaflet [98,99]. A double CARC-CRAC motif has been identified within the transmembrane domains of some membrane proteins [10,98]; however, the limiting factor of the CRAC/CARC sequence is that they are based on a linear (1D) sequence motif, as opposed to cholesterol-binding sites which consist of a three-dimensional (3D) structure [100]. Another common motif is the CCM, which is defined by four spatially distributed interactions with cholesterol: an aromatic Trp158, conserved in 94% of class A GPCRs; a hydrophobic Ile154 conserved in 35% of class A GPCRs (both residues in helix IV); an aromatic Tyr70 from helix II, which forms a hydrogen bond with Arg151 from helix IV [101,102,103]. This motif was established from the analysis of the human β2-adrenergic receptor in a complex with timolol and two molecules of cholesterol [101]. CCM can either be described as strict or less restrictive [101,104]. The strict variant is found in 21% of the class A GPCRs, while the less-restrictive variant, defined by the absence of the aromatic residue from helix II, is present in 44% of class A GPCRs [101,103,104]. Additionally, the sterol-sensing domain is another significant cholesterol recognition motif with a larger protein segment and comprises five transmembrane helices. The sterol-sensing domains usually consist of a tetrapeptide amino acid sequence—tyrosine, isoleucine, tyrosine, and phenylalanine (YIYF)—which has been found to be present in other lipid-raft associated proteins without the SSD motif [90,91,94]. Studies have shown that the presence of the YIYF amino acid sequence alone can interact with the cholesterol-rich domain [90,91,94,105,106]. Finally, START proteins have also been identified as a cholesterol binding motif, with the transport of cholesterol molecules being their primary function [107,108]. Proteins with the START domain [109] are able to transfer lipids between membranes and can interact with cholesterol [94]. While all of these are cholesterol-binding motifs in membrane proteins, the CARC-CRAC motif is the major cholesterol interaction site that has been observed in GPCRs.

### 1.2. GPCR–Cholesterol Interactions

G-protein-coupled receptors are a superfamily of integral membrane proteins in the human genome, constituting one of the largest classes of clinical drug targets [110,111,112,113]. Often distinguished by a characteristic seven transmembrane helices plus an eighth helix that lies underneath the surface of the layer, GPCRs depend on a relationship with the lipid membranes in their physical environment to perform their function [11,62]. As per the phylogenetic investigation, most GPCRs belong to one of four classes, i.e., A, B, C, and Frizzled. The class-C GPCR family contains metabotropic glutamate receptors (mGluR1–8), γ-aminobutyric acid receptors, a few taste-detecting receptors (TAS1R1-3), Ca${}^{2}$${}^{+}$-detecting receptors (CaS), and orphan receptors [114]. One trademark highlight of the class-C GPCRs is their dimerization, either into homo- or hetero-dimers, which is requisite for their proper functioning [40]. Cholesterol assumes an essential role in the function of a significant number of GPCR structures [115]. It does this by binding to a number of GPCRs, including rhodopsin [116], oxytocin [117], μ-Opioid [118], and serotonin 1A receptors [119], at both canonical and non-canonical binding sites, consequently altering their ligand-binding activity allosterically, which could result in the activation or inactivation of the protein. For example, cholesterol is reported to influence Hedgehog (Hh) signaling as a means of activating the Smoothened orphan receptor (SMO) which belongs to GPCRs [120]. As such, it has been determined that cholesterol can influence the stability, oligomerization, and ligand-binding affinity of GPCRs [12,63,116,119,120]. Two mechanisms have been proposed by which cholesterol might influence the structure and function of GPCRs: directly, through specific interactions with the GPCRs; indirectly, by altering the physical properties of the membrane; or perhaps some combination of the two mechanisms [61,92,121,122]. Recently, several GPCR structures have been determined through X-ray diffraction and even more through Cryo-EM. A large percentage of these structures have been stabilized by site-specific cholesterol binding, although it is uncertain if these cholesterol associations are due to recurring cholesterol-binding motifs or if the experimental technique used determines the method of cholesterol binding. A comprehensive study by Taghon et al. [88] showed that cholesterol binding in both X-ray and Cryo-EM structures is much the same. They also indicate that about 92% of cholesterol molecules on GPCR surfaces are located in predictable locations that do not require cholesterol-binding motifs [88]. The importance of cholesterol in GPCR structural dynamics has been identified in some GPCR structures, especially within the class A family (e.g., the presence of CCM in the β2-adrenergic receptor [101]). In some cases, CHS has been used to substitute cholesterol in GPCRs, although the validity of this replacement has been contested [93,123,124,125]. The CRAC motif has been established as a characteristic feature of the serotonin (1A) receptor [126], the β2-adrenergic receptor [127], cholecystokinin [121], cannabinoid (CB1) receptor [128], etc. [92,93,129], indicating that the interaction of cholesterol with GPCRs could be specific in nature. However, another group of researchers suggested that the presence of CRAC/CARC motifs does not automatically prove that cholesterol interacts within those binding motifs [130]. On the flip side, their impact on class C GPCRs is yet to be fully explored [36]. The significance of cholesterol to GPCR structures and their functional dynamics is an ongoing question that is yet to be fully elucidated [93].

## 2. Significance and Interaction Sites of Cholesterol in Class C GPCRs

### 2.1. Metabotropic Glutamate Receptors (mGluRs)

Metabotropic glutamate receptors (mGluRs) are a family of G protein-coupled receptors that are significant in regulating neurotransmission [131,132,133]. GPCRs are membrane-bound proteins expressed in the central nervous system (CNS), and their physiological functions are dependent on their lipid environment [134]. There are three groups with eight subtypes of mGluRs that are classified based on G-protein coupling and ligand selectivity [40,135]. Group I consists of mGluR1 and mGluR5, which are linked to the activation of phospholipase C (PLC) to increase diacylglycerol (DAG) and inositol triphosphate (IP3), Group II includes mGluR2 and 3 [136], Group III is comprised of mGluR4, 6, 7, and 8. These last two groups are linked to the inhibition of adenylyl cyclases (ACs) [137,138]. In mammalian cells, cholesterol is highly concentrated in the plasma membrane but low in the intracellular membrane [139]. Cholesterol affects receptor function by affecting the membrane’s fluidity or interacting with the receptor’s binding site [140] and moves freely between the inner and outer leaflets [141]. In lipid rafts, plasma membranes are rich in cholesterol and sphingolipids within their lipid domains, and the cholesterol forms specific interactions with GPCRs including mGluR1 and mGluR2. Research has shown that cholesterol aids the dimerization of mGluR2 and mGluR5 through interactions with the TM4/TM5 domains and also through the TM1/TM2 domains of mGluR1 [142,143,144]. In mGluRs, investigations from several scientists have suggested specific interaction sites for mGluR1 and mGluR2 [40,145]. A study has revealed the presence of a CRAC motif in the transmembrane helix 5 domain of mGluR1, which is conserved for all mGluRs. The CRAC motif located in TM5 plays an important role in supporting mGluR1 recruitment to the lipid raft as a result of agonist binding [143]. It has been reported that mutations in this motif affect both signaling and the association of mGluR1 with cholesterol-rich membrane domains [143]. Another group has experimentally determined that within the transmembrane domain of mGluR1, cholesterol is localized within the helix I homodimer interface. Intriguingly, this was observed through analyses of the crystal structure of the transmembrane domain of mGluR1, bound by six cholesterol molecules mediating the dimer interface, which in this case is mainly composed of the TM1 helices from both protomers (Figure 2). It has been suggested that these cholesterol molecules stabilize the dimerization of mGluR1 (PDB:4OR2) [36]. In addition, by increasing cholesterol levels, mGluR1 signaling efficiency is enhanced upon stimulation by an agonist, while by lowering cholesterol levels, extracellular signal-regulated kinase-mitogen-activated protein kinase (ERK-MAPK) activation via mGluR1 is inhibited [143,146]. In this way, lipid rafts and membrane cholesterol act as positive allosteric modulators (PAM) of the group I mGluR signaling pathway. Therefore, it is possible to modulate abnormal group I mGluR behavior in neuropsychiatric conditions (fragile X syndrome and autism) through the use of drugs such as statins and cyclodextrins, which affect membrane cholesterol levels [143]. Furthermore, the role of cholesterol has also been considered for class II members of mGluRs. A number of neuropsychiatric conditions, including depression, Alzheimer’s disease, and Parkinson’s disease, as well as different types of cancer, have been treated with these same classes of drugs [147,148,149]. The binding of glutamate to mGluR2 dimers results in the transmission of a signal across the transmembrane domain of the receptor that prevents the activity of adenylate cyclase via the G${}_{i/o}$ protein [135]. In a study, the interaction of cholesterol with mGluR2 was demonstrated across 2 to 5 sites in the transmembrane domain of mGluR2 [135], using molecular dynamics simulations [150,151,152], biochemical approaches, and photocrosslinking experiments. It was observed that mGluR2 is modulated by their surrounding lipid environment, particularly cholesterol, through an unknown mechanism. The CRAC/CARC motif and a cholesterol consensus motif (CCM) were suggested as cholesterol-binding motifs in GPCRs [67,153]. A central aromatic amino acid that interacts with sterols is a common characteristic of some of the motifs described [135]. A recent computational study conducted by Bruno et al. [154] found that the conformational differences observed in the helical structure of the mGluR2-TM8 domain can be used as an indicator to detect the presence of cholesterol in metabotropic glutamate receptors and GPCRs. They observed that the inclusion of higher levels of cholesterol in the membrane stabilizes the transmembrane helix 8 (TM8) of mGluR2, while a lack of cholesterol results in destabilization of the TM8 domain [154]. However, the role of cholesterol in the third group of mGluRs remains unknown.

### 2.2. GABAB Receptors

In mammals, GABA (γ-Aminobutyric acid) is one of the major inhibitory neurotransmitters. In order for GABA to exert its effects, it must bind to at least two different receptor classes: GABA${}_{A}$ and GABA${}_{B}$. Approximately 20 to 50% of the brain’s synapses contain GABA${}_{A}$ receptors [155]. They are pentameric receptors belonging to a superfamily of ligand-gated ion channels [29]. Unlike GABA${}_{A}$, GABA${}_{B}$ receptors are members of class C GPCRs with the typical classification of an N-terminal VFT region: a 7TM domain, and a C-terminal intracellular domain [24,30,156]. GABA${}_{B}$ receptors function as inhibitor receptors by opening potassium channels, reducing the activity of adenylate cyclase and calcium channels [157]. There are few solved structures of GABA${}_{B}$ receptors containing cholesterol, deposited on the protein data bank (Figure 3), and subsequently, there is little knowledge of the effect of membrane cholesterol on the GABA${}_{B}$ receptors. Experimental investigations have shown that cholesterol enrichment and depletion both decrease GABA potency, resulting in an up to fourfold increase in EC50 [158]. The structures of GABA${}_{B}$ receptors with cholesterol were determined based on ligand type because the presence of a ligand can change how cholesterol interacts with the receptor. For instance, in absence of ligands, it is feasible that there is no interaction between cholesterol and the receptors. However, for systems bound to an antagonist, 10 and 16 molecules of cholesterol [24,65] were bound between the protomers of the transmembrane dimers [115] (Figure 3). Thus, It can be suggested that the ligands may have caused some conformational changes in receptors allowing greater binding to cholesterol. Moreover, three cholesterol molecules were attached to the GABA${}_{B}$ receptor bound to a positive allosteric modulator (PAM) [65] (Figure 3). Therefore, the variation in cholesterol binding between two different ligand classes can indicate the potentially significant role of ligands in the interaction between cholesterol and the receptors.

### 2.3. Taste Receptor

TAS1R1 and TAS1R2 were among the first determined subfamilies of taste-related GPCRs. Prior to identifying their physiological ligands, they were originally classified as orphan receptors [159]. Subsequently, some scientists identified a member, TAS1R3, through a fusion of molecular biological and genetic approaches [160]. These three members (TAS1R1-3) code for sweet and umami tastes and are classified as class C GPCRs. The sweet taste signals are activated by TAS1R2 and TAS1R3 heterodimers, while the umami taste signals are transduced by heterodimers of TAS1R1 and TAS1R3 [161]. Therefore, the class C taste receptors consist of either TAS1R1 or TAS1R2, interacting with a common subunit TAS1R3. Similar to other class C GPCRs, they exist as obligate dimers and are characterized by a large extracellular N-terminus, which houses the orthosteric ligand-binding site, while the allosteric binding sites are present in the cysteine-rich domain and/or transmembrane region [160]. Due to these multiple binding sites, a single taste receptor is able to function for various stimuli [53]. The sweet taste receptor is able to interact with various compounds at a lower sensitivity, unlike most GPCRs, which are highly selective to specific high-affinity ligands. Cholesterol has been shown to regulate GPCR signaling in sweet taste receptors [162,163]. A study showing the presence of a CRAC motif in T2R4 (a subset of GPCRs responsible for bitter taste receptors [164]), explains that taste receptors are crucial to cholesterol sensitivity [94] and become more sensitive to cholesterol through a cellular mechanism [92]. Furthermore, they observe electrostatic interactions between the 3β-hydroxyl group of cholesterol and the positively charged residue in the cholesterol binding motif [162]. Site-directed mutagenesis and functional assays have been optimized in the study of putative cholesterol-binding motifs (CRAC and CARC) to determine the mechanism of cholesterol binding to taste receptors. A comparison of the dynamics of wild-type T2R14 receptors and mutant T2R14 receptors revealed that the amino acid residues K110, F236, and L239 are required for the receptor to function appropriately when cholesterol is present. Based on this study, it could be suggested that cholesterol influences taste receptors by directly interacting with the receptor [165].

### 2.4. Retinoic Acid-Inducible Orphan G Protein-Coupled Receptors (RAIGs)

Retinoic-acid inducible receptors belong to a group of class C GPCRs [166]. Although containing a characteristic secondary structure of seven transmembrane α-helical domains, these receptors have short amino-terminal extracellular domains, ranging from 30 to 50 amino acids [167]. In contrast, other family C members consist of a large N-terminal domain, comprising 500–600 amino acids [168]. Currently, there are four genes that make up the RAIG family: RAIG1, RAIG2 RAIG3, and GPCR5D [50]. RAIG1 was the first of these genes to be determined, and it was associated with a retinoic acid-responsive gene in human carcinoma cells [50,167,168]. Although classified as a class C GPCR, RAIG protein shares low sequence similarity with known members of GPCRs, and only shows 25% similarity [167,169] with the homology sequence of mGluR2 and 3, primarily in the transmembrane regions [167]. Due to the large variation between the sequence homology of RAIG proteins and most GPCRs, the endogenous ligands for RAIGs remain unknown [169]. However, the ligand-binding regions are predicted to be found in the extracellular loops of the transmembrane domain, and also at the short amino-terminal regions [167,168,170]. A study that utilized fluorescence microscopy and immunocytochemical methods to study the formation and localization of synaptic vesicles in human SH-SY5Y neuroblastoma cells suggested that retinoic acid-induced proteins with cholesterol produced significant neurite extension and formation of cell-to-cell contacts, predicting it as a valuable tool for basic studies of neuronal metabolism [166].

### 2.5. Calcium-Sensing Receptor-Related Receptor

As a G-protein-coupled receptor, the calcium-sensing receptor (CaSR) is essential for controlling calcium homeostasis [171] in humans. CaSR is a Ca${}^{2}$${}^{+}$-sensing protein found on the surface of cells [172] that exists as an obligate homodimer and belongs to class C GPCRs [48]. Each protomer has a Ca${}^{2}$${}^{+}$-binding extracellular domain and a seven-transmembrane-helix domain (7TM) that activates heterotrimeric G-proteins [34]. The classical calcium-sensing receptor is known to be involved in the pathophysiology of parathyroid and renal-related diseases by sensing calcium ions in extracellular fluid [53,171]. Recent studies suggest that CaSR can be modulated through the interactions of its transmembrane (TM) domains with cholesterol [66]. This is especially observed in TM6 where cholesterol molecules found at the dimer interface influence the interactions of two residues (ILE816) from the TM6 helices of both subunits. This allows the side chains of both residues to pack against each other and make indirect dimer contacts. As a result of these observations, it could be suggested that the TM6-TM6 dimer interface is stabilized by cholesterol and as such it is essential in the receptor activation of CaSR [66].

Cholesterol depletion has been shown to negatively impact receptor function by decreasing basal activity and Ca${}^{2}$${}^{+}$ sensitivity [173]. Another study showed that vascular smooth muscle cells (VSMCs) [174] are expressed in CaSR and can be altered by cholesterol [175]. They further indicated that plaque stability can be affected due to CaSR [176] mediating MMP-2 (matrix metalloproteinase-2) production in the presence of cholesterol via the phosphatidylinositol 3-kinase (PI3K)/Akt signal pathway [177,178]. In addition, activation of CaSR in VSMCs increases cell proliferation and survival via the phospholipase C (PLC)-IP3 and MAPK-ERK1/2 pathways [179].

### 2.6. Orphan Receptor

Despite extensive reorganization efforts, there are over 140 receptors [180] within the GPCR family that have yet to be fully identified and these groups are referred to as orphan receptors [37,112,180,181]. Orphan GPCRs play important roles in physiology and diseases, yet they are poorly understood in terms of their structural organization, ligand identification, activation mechanisms, and signaling reactions [56,180,182] GPR156, GPR158, and GPR179 make up the orphan receptor of class C GPCRs [56,57,58,59,60,182] and they are the least characterized members of the group [182]. They share 70% sequence similarity in both extracellular and TM domains, with a distinct feature of lacking the Venus flytrap-fold ligand-binding domain [182]. GPR158 are drug targets with significant roles in mood regulation, memory, depression [183], carcinogenesis, and cognition [56,182,184,185,186]. It is highly expressed in brain tissues [187] and functions by regulating ion channels and second messengers. One prominent characteristic of GPR158 is that it binds to the neuronal RGS7-Gβ5, a regulator of the G protein signaling protein complex [188], that directly deactivates G proteins [188,189]. GPR158 and RGS regulate the homeostasis of the second messenger cyclic adenosine monophosphate (cAMP), and control the neuronal activity with a marked impact on brain physiology [56]. Recently, two high-resolution Cryo-EM structures have been determined by Patil et al. [56]. The structures consist of GPR158 alone, and GPR158 bound to RGS complex (Figure 4), with both determined structures revealing the presence of an extracellular Cache domain and an unusual ligand-binding domain, that is not found in other GPCRs [56]. In both initial structures, cholesterol interacts between the protomers and the transmembrane helices to stabilize the protomers [56]. The determined structures show several cholesterol molecules surrounding the dimeric interface of GPR158, which acts as a shield for the cavity formed at the interface. It is also suggested that the interactions of cholesterol with the transmembrane helices could stabilize the interface between both protomers. Patil et al. [56] reported that the stability provided by these cholesterol molecules results in a more compact dimeric interface, which then prevents G protein activation [56].

## 3. Conclusions

Through this review and the accompanying table and figures, we have described the interaction sites of cholesterol in specific receptors of class C GPCR structures. Through the collective study of class C GPCR structures, we notice that cholesterol is mostly bound between the transmembrane dimers of the receptors and also within the surrounding groves of the transmembrane helices, which could explain why it seems to aid dimerization. Furthermore, this review highlights the significance of cholesterol within specific class C GPCRs. Consideration of several studies revealed that cholesterol is important for oligomerization, organization, function, and dynamics of class C GPCRs. In general, we see that cholesterol could affect ligand binding, G-protein coupling, and intracellular signaling of GPCRs. With the possible emergence of more cholesterol-bound GPCR structures and analyses, we picture an exciting and enlightening future in the study of cholesterol–GPCR interactions. We expect that this information will help provide insight into the molecular mechanisms of cholesterol molecules bound to particular receptors of class C GPCRs.

## Figures and Tables

**Figure 1 membranes-13-00301-f001:**
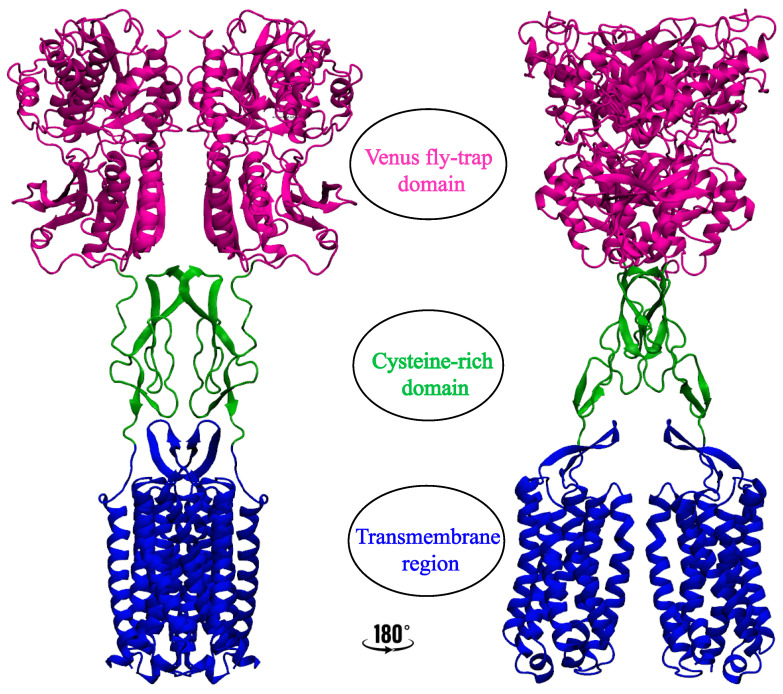
Representation of a class C GPCR (a full length human mGluR5) showing the different regions as: VFT (magenta), CRD (green), and 7TM region (blue). (PDB ID: 7FD8).

**Figure 2 membranes-13-00301-f002:**
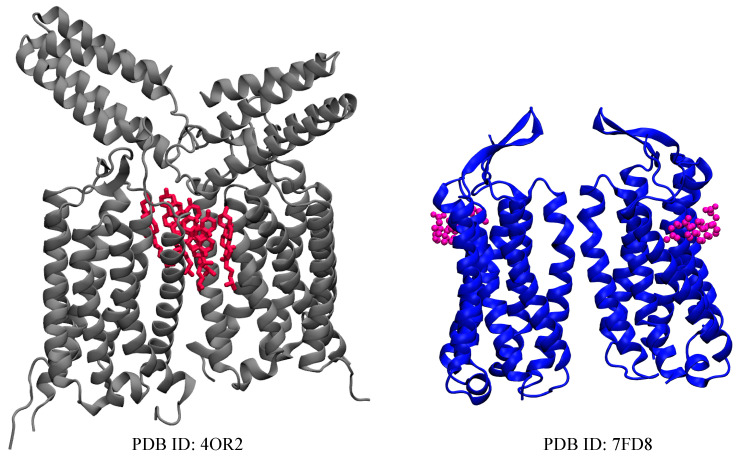
Crystal structure of mGluR1 (grey) in an inactive state, and a Cryo-EM structure of an intermediate-active mGluR5 (blue), determined with 6 molecules of cholesterol (red) and 2 molecules of CHS (magenta), respectively.

**Figure 3 membranes-13-00301-f003:**
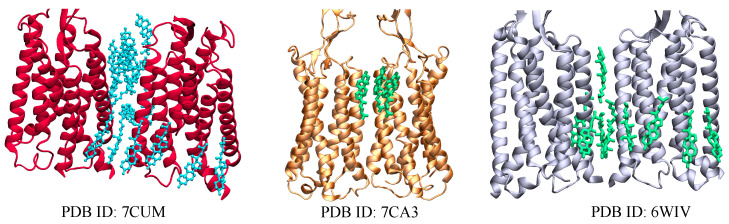
Visual representations of GABA${}_{B}$ receptors in active state (orange) and inactive states (red and gray), determined by single particle Cryo-EM. They include 2 (orange), 17 (red), and 16 (gray) bound cholesterols, respectively, within the transmembrane region. The figures here show the transmembrane region only bound to cholesterol (cholesterol molecules are shown as cyan and green sticks).

**Figure 4 membranes-13-00301-f004:**
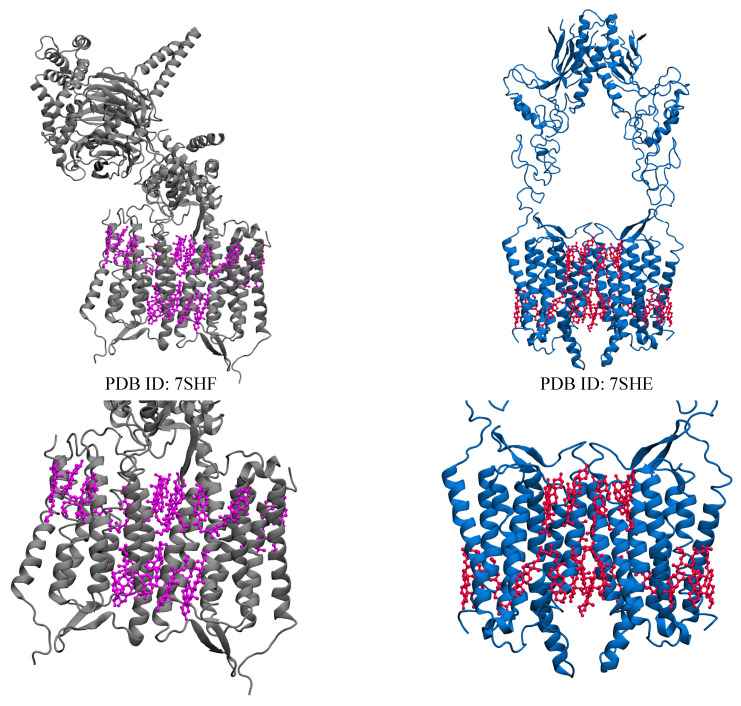
Visual representations of inactive states of GPR158 class C orphan receptors, showing GPR158 bound to RGS complex (gray) and GPR158 in apo form (blue). Both structures contain 22 cholesterol molecules within the transmembrane region. The upper figures show the whole protein, while the lower figures only show the transmembrane region (cholesterol molecules are shown as violet and red sticks).

**Table 1 membranes-13-00301-t001:** Solved class C GPCR structures with cholesterol acting as ligands.

Name	PDB ID *	Number of Sterols in TM
mGluR1	4OR2 [36]	6 CLR
mGluR5	7FD8 [45]	2 CHS
GABA${}_{B}$	6WIV [24]	10 CLR
	7CUM [65]	16 CLR
	7CA3 [65]	3 CLR
CaSR	7SIM [66]	8 CLR
	7SIL [66]	8 CLR
Orphan receptor (GPR158)	7SHF [56]	22 CLR
	7SHE [56]	22 CLR **

* Data obtained from protein data bank (PDB) database (https://www.rcsb.org, accessed on 9 February 2023). Citations to the PDB structures are included. ** CLR cholesterol; CHS cholesteryl hemisuccinate.

## Data Availability

No new data is reported in this review article.

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
