# Peer review of "Cholesterol in Class C GPCRs: Role, Relevance, and Localization"

_membranes, 2023, doi:10.3390/membranes13030301_

Round 1
Reviewer 1 Report
The authors submitted a review of the interaction of cholesterol molecules with members of class C of the GPCR family. They subdivided the manuscript by subclasses and gathered structural and functional results related to cholesterol impact and interaction with members of this GPCR's class.
Some major and minor comments could help to improve the manuscript.
Major comments
- Add a section about members of the RAIG's subfamily
- Delete text about nuclear receptors from lines 314 to 335, and 339 to 344.
- Provide more information about GRP158 and the role of cholesterol for this GPCR that had one the highest number (22) of cholesterol molecules bound in the structures.
- Line 58: The table 1 does not indicate cholesterol molecules as "ligand" but as interacting molecules in structures
- Table 1: update the table with missing structures having cholesterol: 5 CaS, 1 GABAB is missing, 1 GPR 158 is missing.
- Line 94: indicate the other cholesterol binding motif CCM.
-
Minor comments:
- Line 29: "other responsibilities for lipids and sterols". It is not clear what could be the other responsabilities. Please clarify.
- Line 51: indicate that the 20 structures are from human class C GPCRs.
- Line 58: structures have been obtained also by cryo-EM
- Move table 1 in line 64.
- Line 70: "… viability and cell proliferation." . Add reference.
- Line 78: "… lipid membranes in the cell.". Add reference.
- Line 136: delete "structures".
- Line 136: add the GPCRs cited in line 144 and delete the related sentence in line 144.
- Line 139: " dependability" is not a function. Delete or clarify.
- Table 1: title of last column: " Choleseterols": replace by "sterols"
- Under Table 1: " CHS cholesterol hemisuccinate", replace by " CHS cholesteryl hemisuccinate"
- Fig. 1: indicate PDB in legend
- Line 182: " included in the membrane of mGluRs". mGluRs do not have membrane.
- Line 247: The number of molecules does not fit with the Fig. 3.
- Fig. 3: indicate the state of GPCR above structures
- Line 258: replace "Linked" by "interacting".
- Fig. 4: dived in panels. Explain the differences between the grey and blue structures.
Author Response
Please see the files attached.

Reviewer 2 Report
This is a revised manuscript that was originaly submitted to CellChemBiol. As such, the authors completed the revision according to my original critocisms, and I m pleased to say that I am fully staisfied by their responses, the additional data they provided and their modification in the text.
Author Response
Please see the file attached.

Reviewer 3 Report
This report reviews the role of cholesterol specific to Class C GPCRs. The review covers a structural analysis of existing PDB structures of Class C GPCRs and maps the location of cholesterol and cholesterol binding sites in the proteins.
Where the paper could be improved is in providing some thoughtful insight on what roles these interactions play in the mechanism of these receptors and any identifying features specific to the class.
Author Response
Please see the files attached.

Round 2
Reviewer 1 Report
The authors responded to all comments made in the initial manuscript.
No new comments are raised and I suggest the publication of the revised manuscript.
For information concerning a previous comment about additional CaS' structures with cholesterol, authors are right that only 2 structures contains cholesterol molecules in the PDBs. However, in the manuscripts of 3 others structures (7M3F, 7M3G, 7E6U), densities of cholesterol molecules are observed but not included in the PDBs.